# Sex-Specific and Long-Term Impacts of Early-Life Venlafaxine Exposure in Zebrafish

**DOI:** 10.3390/biology11020250

**Published:** 2022-02-06

**Authors:** William Andrew Thompson, Zachary Shvartsburd, Mathilakath M. Vijayan

**Affiliations:** Department of Biological Sciences, University of Calgary, Calgary, AB T2N 1N4, Canada; william.thompson@ucalgary.ca (W.A.T.); zachary.shvartsbu1@ucalgary.ca (Z.S.)

**Keywords:** municipal wastewater effluent, behaviour, metabolism, swimming, antidepressants, serotonin, catecholamines, stress response, anxiety, activity

## Abstract

**Simple Summary:**

Excessive use of antidepressants, combined with our inability to completely clear them from municipal wastewater effluents, has led to their increased presence in aquatic habitats. Venlafaxine, one of the more commonly prescribed antidepressants, has been shown to be detrimental to the early life stages of non-target animals such as fish. Exposure to venlafaxine at the embryonic stage appears to lead to behavioural disruptions when zebrafish become free swimming and reduces growth in juveniles. Here we tested whether early-life exposure also led to behavioural and metabolic perturbations in adults using zebrafish, a widely utilized model in developmental toxicology. Zygotic exposure to venlafaxine compromised activity and anxiety responses and reduced the active metabolic rate as well as the aerobic scope in a sex-specific manner. This study raises the possibility that early developmental exposure to venlafaxine may have long-term consequences on fish performance, and that this may be sex dependent.

**Abstract:**

Venlafaxine, a selective serotonin and norepinephrine reuptake inhibitor, is a widely prescribed antidepressant that is detected in municipal wastewater effluents at µg/L concentrations. It has been shown to impact the early life stages of fish, including neurodevelopment and behaviour in larvae, but whether such early exposures have longer-term consequences are far from clear. Here, we sought to determine whether zygotic deposition of venlafaxine, mimicking a maternal transfer scenario, disturbs the metabolic rate and behavioural performance using zebrafish (*Danio rerio*). This was tested using freshly fertilized embryos (1–4 cell stage) microinjected with either 0, 1 or 10 ng of venlafaxine and raised to either juvenile (60 days post-fertilization) or adult (10–12 months post-fertilization). Zygotic venlafaxine exposure led to a reduction in the active metabolic rate and aerobic scope, but this was only observed in female fish. On the other hand, the total distance travelled in an open field assessment was greater at the highest concentration of venlafaxine only in the adult males. At the juvenile stage, behavioural assessments demonstrated that venlafaxine exposure may increase boldness—including hyperactivity, lower thigmotaxis, and a reduction in the distance to a novel object. Taken together, these results demonstrate that zygotic venlafaxine exposure may impact developmental programming in a sex-specific manner in fish.

## 1. Introduction

Venlafaxine, a selective serotonin and norepinephrine reuptake inhibitor [1] is commonly found at levels exceeding 300 ng/L environmentally [2,3,4]. This raises concerns over possible impacts of this pharmaceutical on non-target organisms, including fish, in the aquatic habitats receiving municipal wastewater effluents [5]. Studies have shown that early-life exposure to venlafaxine disrupts developmental programming, including altered neurodevelopment, cardiac development, behaviour, and growth [6,7,8,9,10,11,12]. Additionally, we recently showed that early life exposure may have long-term consequences on the growth and stress performance of adult zebrafish [10,13]. While early-life exposure to venlafaxine diminishes larval behavioural performance [6,7,9], still unknown are whether these behavioural disturbances are persistent in adulthood. Moreover, considering the backdrop of long-term changes in growth and stress responses [10,13], it is possible that the metabolic performance of these animals are also affected, but this has yet to be tested.

Behavioural responses are the summation of cognitive, morphological, and physiological responses to abiotic and biotic factors in the environment [14]. Several studies have linked monoaminergic systems as key players in regulating behavioural profiles in animals [15,16,17]. Behavioural responses, in turn, have been used to forecast responses to predators or adverse situations in fish [18,19]. Assessments of behaviour can be complex, particularly at the adult stage, where spatial and exploratory behaviours are more robust [20,21,22]. The usage of novelty, by introducing animals to either a new arena or object, to assess activity, anxiety, and fear is necessary to characterize behaviour and spatial strategies [22]. For example, measurement of thigmotaxis has been used extensively in fish as a measure of anxiety-like behaviour, as teleosts exhibit a conserved behavioural response of spending time exploring either the bottom or the periphery of the arena when anxious [21,23,24]. We have previously described venlafaxine-mediated impacts on larval behaviour in response to zygotic and water-borne exposures, linking these outcomes to disruptions in neurogenesis, serotonin expression, and catecholamine cell populations in zebrafish [6,7,10]. Studies extending the impact of early-life exposure have suggested life-long effects on the endocrine regulation of growth [13] and the stress response [10]. Interestingly, our results suggest that this early-life exposure impact on the endocrine stress response is sex-specific at the adult stage [10].

In this study we tested the hypothesis that early-life venlafaxine exposure will compromise the long-term behavioural and metabolic performance of fish. Using zebrafish as a model, we mimicked a maternal venlafaxine transfer scenario by injecting either 0, 1 or 10 ng of this antidepressant into embryos at the 1–4 cell stage. We have previously shown that this mode of exposure results in developmental changes in zebrafish that are long-lasting, including disruptions to juvenile growth and adult stress responses, and reflects effects on developmental programming [7,9,10,13]. In this study, we specifically asked the question of whether venlafaxine impacts on metabolism and behaviour are sex-specific in zebrafish. This was carried out by measuring the metabolic rate, swimming performance, and behavioural performance of adult zebrafish.

## 2. Materials and Methods

### 2.1. Fish Husbandry and Microinjection

Lab populations of Tupfel long-fin strain (TL) adult zebrafish were maintained at 28.5 °C, pH ~7.9, ~820 µs conductivity, 14: 10 light: dark cycle on aquatic habitat recirculating systems (Pentair, Minneapolis, MN, USA), and fed twice daily. Zebrafish were moved into breeding traps in equal proportions of males and females following their second feed and held overnight. Eggs were collected immediately after fertilization and were injected (at the 1–4 cell stage) with a Narshige IM 300 micro-injector under a Leica M165 dissecting microscope (Leica Microsystems, Wetzlar, Germany). Prior to injection, the microinjector was calibrated by calculating volume against a stage micrometer submerged in mineral oil. A total of 0, 1 and 10 ng of venlafaxine (at a total of 1 nL; dissolved in dd H_2_O; CAS # 99300-78-4; Sigma-Aldrich, St. Louis, MA, USA) were injected into each embryo, with phenol red used to confirm injection into the yolk sac (final concentration 0.0025%), as described previously [7]. Additionally, the 1 or 10 ng concentration of venlafaxine injected into the zygote provided a dosage of 0.4 and 4.2 ng per embryo, respectively [7]. Following injection, embryos were raised until 5 days post-fertilization (dpf) in the same light cycle as adults; following which, zebrafish were raised in static 3 L tanks (n~30) with system water held at 28.5 °C and fed gemma twice daily until 15 dpf. After this, the 3 L tanks were moved onto the recirculating systems described above and were fed twice daily with gemma food in the mornings and brine shrimp in the afternoon until experimentation. Twelve independent clutches of embryos from unexposed breeders were collected, approximately 300 eggs per clutch, and microinjected with all treatment concentrations (~100 eggs per treatment). Treatments were raised separately in 3 L tanks in groups of ~20 fish, and we have previously shown that these exposures did not affect their survival [7,13]. The experimental design and the sampling times are shown in Figure 1. All metabolic and behavioural trials were carried out with different set of fish. Only the open field and the subsequent novel object assessments were carried out with the same fish. There were no changes in the length and body mass of either male or female adult fish between treatments used in this study.

### 2.2. Metabolism and U_crit_

Adult zebrafish (10–12 months post-fertilization) were separated by sex, weighed, and placed in 45 mm respirometers (Loligo systems, Viborg, Denmark) submerged in a water bath maintained at 28 °C. Zebrafish were allowed to acclimate in this chamber for 24 h. The following day, zebrafish were assessed for standard metabolic rate over 3 h. Following which, zebrafish were removed from the respirometer and subjected to multiple stressors, including gently chasing the fish in a 2 L container with 500 mL system water for 1 min, followed by air exposure for 1 min. Post-stressor, the zebrafish were returned to the respirometer chamber and immediately assessed for active metabolic rate for 1 h. Metabolic rate was recorded in 8 min intervals, consisting of 5 min flushing periods, a 1 min wait period, and a 2 min closed respirometry window. Data were collected using the Auto Resp software (Loligo systems, Viborg, Denmark), by taking into account the mass of each fish. Values for standard metabolic rate were selected by selecting the 3 lowest consecutive data points during the 3 h measurement period. Active metabolic rate was calculated as the singular highest data point in the 1 h test period. A corresponding blank chamber was run with every replicate to subtract the bacterial respiration from the corresponding value at that time point. Coefficient of determination (R^2^) values higher than 0.9 were used as described previously [25]. A total of 4 respirometers were used concurrently, with one used for each treatment, and one kept blank for bacterial respiration recordings. Fish were taken from 3 replicate tanks for each treatment (total *n* = 9–12), with male or female fish selected randomly from each treatment tank, and following assessment placed in a new tank to prevent them from being reused in the trial.

Swimming performance was assessed using a Loligo 5 L swim tunnel (#SW10050; 230 V/50 Hz), with minor adjustments made to adapt for zebrafish. The perforated plastic sheeting at the start of the swim chamber was removed and replaced with a stainless-steel gradient to prevent zebrafish from finding small pockets of limited flow for recovery. We utilized the U_crit_ protocol established by Wakamatsu et al. [26], with minor modifications. Adult zebrafish had food withheld for 24 h and were then placed into the swim tunnel held at 28 °C. Initial flow rate in the swim tunnel was 10 cm/s for a 1 min adjustment period, followed by a 1 min warm-up period at 15 cm/s. Speed was increased to 20 cm/s, then increased at 1 cm/s every 1 min until the animal could no longer maintain a normal swimming position and collapsed against the back mesh. If the zebrafish could recover within 3 s after collapsing the trial would continue. Following the completion of the trial, zebrafish were removed from the swim tunnel and length and weight taken. Calibrations of swimming speed was completed every 3 d using a digital anemometer (Hontzsch (AC 10000) and a 30 mm vane wheel flow probe (AC 10002). The swimming protocol was carried out using the Loligo AutoResp software, and U_Crit_ was calculated as described previously [27].

### 2.3. Behaviour

Zebrafish were assessed for behavioural changes, including vertical swimming behaviour (lateral view), open field (above view), and novel object (above view) assessments. These assessments were chosen to provide multiple angles for assessing thigmotaxis and activity [21,24]. All assessments were recorded with a Sony Handycam CX405 camcorder (Tokyo, Japan). Lateral view assessments were carried out by placing zebrafish into 2 L of system water in a container measuring 15 cm × 15 cm × 20 cm. The tank was placed onto a white base, and the digital camera placed approximately 50 cm horizontal from the arena (Figure 1D). Top view assessments (open field/novel object) were carried out inside a white container (80 cm × 80 cm × 25 cm) with two PVC pipes affixed to the edges to hold the camera above the chamber (Figure 1B,C). Zebrafish were placed in a total of 1 L of system water in a 15 cm × 15 cm × 20 cm tank, which was placed in the centre of the white container described above. The open field assessment occurred for a total of 5 min. Following which, filming was paused to place a novel object (a cylindrical air stone; 2 cm × 4 cm) inside the centre of the arena (Figure 1C), and filming resumed for 5 min with the same fish used in the open field assessment. Juvenile (total *n* = 9–12), male (*n* = 6–16) and female (*n* = 6–17) adult zebrafish underwent these behavioural assessments. For juveniles, two replicate tanks were used for behavioural assessments, with each tank contributing 6 fish that were randomly selected for each treatment, and fish placed in a recovery tank to prevent their reuse. Adult zebrafish assessments for the vertical assessment or the open field/novel object assessment were carried out using all the fish in those 2 tanks from each treatment. As entire tanks were used for this assessment, sex ratios between treatments were random and noted after the trial. Individuals were removed from the assessment if they failed to move throughout the duration of the trial period. Noldus Ethovision XT 14 was used to analyse all video files, tracking animals for total distance travelled and relative position in the tank for thigmotaxis. To assess thigmotaxis, we created distinct “bold” areas within each arena. For vertical assessments, 4 equal blocks were made in the arena, with the uppermost block considered the region of “boldness”. For the open field assessment, the rectangular arena was divided into 9 equal blocks (3 × 3), with the centre block used as our region of “boldness”. The novel object itself was pinpointed in our assessment, and the total distance away from the object was tallied. Thigmotaxis (%) was calculated as (time outside/total time × 100).

### 2.4. Statistics

Data are shown as individual data points with mean ± S.E.M. All figures were analysed via one-way ANOVA, and in the case of a significant main effect, a Tukey post-hoc test was carried out for comparisons between groups. Thigmotaxis data underwent arc-sin transformation prior to analysis, and untransformed data is shown in the figures. In the case of violations of normality or homoscedasticity, data sets were transformed. For all analyses, a significance of 0.05 was applied, with all data analysed using Prism 7.0 software (Graphpad, CA, USA).

## 3. Results

### 3.1. Metabolism and Swimming Performance

There were no discernible differences in the standard metabolic rate of adult female zebrafish following venlafaxine deposition (Figure 2A; *p* > 0.05). Zygotic venlafaxine exposure had a significant effect on the metabolic rate following a stressor (*p* = 0.011; Figure 2B). The active metabolic rate significantly dropped to 1063 ± 102.6 mgO_2_/kg/h in 1 ng venlafaxine fish (*p* = 0.04) and 993.6 ± 101.2 mgO_2_/kg/h in 10 ng venlafaxine fish (*p* = 0.01) when compared to the control fish (1429 ± 89.9 mgO_2_/kg/h). The overall aerobic scope was also impacted by zygotic venlafaxine deposition (*p* = 0.005; Figure 2C). Control fish exhibited an aerobic scope of 1095 ± 85.4 mgO_2_/kg/h, which dropped to 731.6 ± 109.5 mgO_2_/kg/h in 1 ng venlafaxine-treated fish (*p* = 0.03), and to 623.6 ± 88.22 mgO_2_/kg/h in 10 ng-treated fish (*p* = 0.005). Venlafaxine deposition did not significantly affect the standard or active metabolic rate of male zebrafish (*p* > 0.05; Figure 2D–F). Additionally, venlafaxine deposition did not affect the U_Crit_ of zebrafish, regardless of sex (*p* > 0.05; Appendix A).

### 3.2. Behaviour

#### 3.2.1. Juvenile Stage

In order to assess animal exploration strategies in novel environments, we employed several different assessments for anxiety, activity, and aversion behaviours [20,21,22]. Venlafaxine deposition did not alter the vertical swimming behaviour of zebrafish juveniles (Figure 3A,B; *p* > 0.05). However, in the open-field assessment, venlafaxine treatment showed a significant effect (Figure 3C; *p* = 0.023)—the total distance travelled was higher at 1 ng but not 10 ng venlafaxine-treated fish (*p* = 0.018) compared to the control fish. Anxiety-like behaviour, or thigmotaxis, during this measurement was also influenced by venlafaxine deposition (Figure 3D; *p* = 0.034). Venlafaxine reduced the thigmotaxis score of 10 ng-treated fish relative to control fish (*p* = 0.039). The antidepressant also appeared to influence fear responses in juvenile zebrafish (Figure 3E; *p* = 0.048), as 1 ng venlafaxine-treated fish were less aversive to a novel object than control fish (*p* = 0.048).

#### 3.2.2. Adult Stage

Female fish exposed to venlafaxine did not exhibit any differences in thigmotaxis or total distance travelled in any of the behavioural paradigms assessed here (*p* > 0.05; Figure 4A–C; Appendix A). The deposition of venlafaxine did significantly affect activity in male zebrafish (*p* = 0.028; Figure 4D); male fish raised from 10 ng-exposed zygotes travelled a greater distance than controls in the vertical swimming assessment (*p* = 0.025). There was no significant effect of venlafaxine on the behaviour of males in the thigmotaxic responses during the vertical, open field, or novel object assessments (*p* > 0.05; Figure 4E,F; Appendix A).

## 4. Discussion

Our results demonstrate the capacity of venlafaxine to program long-term metabolic and behavioural performances following zygotic deposition in zebrafish. We have shown previously that the zygotic route of exposure, which mimics the maternal transfer of this contaminant, disrupts neurodevelopment and impacts larval behaviour [7,9]. This may be primarily mediated by disrupting the development of the early serotonergic and catecholaminergic systems [9], as the action of venlafaxine inhibits the reuptake of these monoamines in humans [1]. Here, we show that early developmental exposure to venlafaxine has effects that are sex-specific, including reduced aerobic scope in females and hyperactivity in males. Given that zygotic venlafaxine deposition also attenuates the cortisol response to stress—but only in female fish [10]—this raises the possibility that early-life exposure to venlafaxine programs developmental changes in a sex-specific manner in zebrafish. In an environmental context, the impact of venlafaxine may be detrimental to the animal, as reduced aerobic scope and perturbations in locomotory activity are indicative of reduced fitness in fish [19,28].

We have previously reported alterations in early larval behaviour following zygotic venlafaxine deposition [7,9] but had not explored whether similar disruptions are observable as the animal develops. In this study, we show that early developmental venlafaxine exposure has long-term effects, including reduced anxiety-like behaviour and increased boldness as juveniles, followed by sex-specific hyperactive behaviour as adults. We have previously investigated whether thigmotaxis would be affected by venlafaxine in larval fish using two models, the zebrafish and the fathead minnow (*Pimephales promelas*). In both studies, early-life exposure to venlafaxine does not appear to alter larval thigmotaxis [6,7]. Here, we show that zygotic exposure to venlafaxine affects long-term anxiety-like behaviour, including reduced time spent at the edge of the open field assessment and moving closer to a novel object, in juvenile zebrafish. Combined, these results demonstrate that early-life venlafaxine exposure can program increased boldness in juvenile fish. Changes in boldness may have large implications in the survival of animals, as increased anxiety has been linked to elevated shelter behaviours [29], and increased boldness has been directly linked to elevated predation in roach (*Rutilus rutilus*) compared to their more risk aversive conspecifics [30]. Venlafaxine exposure at the zygotic stage appears to program sexually dimorphic effects in adult zebrafish, with males exhibiting changes in hyperactivity—but the mechanisms behind this remain unclear.

One possible explanation for the sex-specific changes in behavioural performance may be associated with alterations in monoamine content. For instance, serotonin, a major target of antidepressant action, is a key monoamine modulating fish behaviour [31], and disruptions in serotonin dynamics impact behavioural responses [32,33,34]. We showed previously that venlafaxine deposition in zebrafish embryos disrupts the early development of the monoaminergic system, including reduced serotonergic immunoreactivity and a decrease in the catecholaminergic cell populations of the embryonic zebrafish brain [9]. Interestingly, these changes, along with the associated behavioural changes, were rescued by serotonin and isoproterenol (a β-adrenergic agonist) treatment, supporting a role for zygotic venlafaxine deposition in interfering with the normal development of the monoaminergic system [9]. This monoaminergic system, including serotonin synthesis and release, is sexually dimorphic [35,36], and correlates with the sex-specific behavioural responses in mammals [37,38]. For instance, changes in serotonin content have been related to aggression and boldness behaviours in zebrafish males, with male and female zebrafish exhibiting regional differences in transcript abundances of serotonin receptors (reviewed by [31]). Additionally, studies have suggested that serotonin transporter expression is sexually dimorphic in vertebrates [39], with serotonin receptor 1_A_ being modulated by sex steroids [40]. Behavioural responses have been shown to be sexually dimorphic in rats, with sex-specific effects mediated via the serotonin 1_A_ receptor [41]. In fish, a study observing brown ghost knifefish (*Apteronotus leptorhynchus*) demonstrated that serotonin content in the central posterior nucleus is sexually divergent, and that serotonin levels correlate with complex communication behaviours, such as reproductive electrocommunication signals, in males [42]. The behavioural changes observed in the present study suggests a preponderance for boldness and hyperactivity, leading to the proposal that disruptions in the developmental programming of the serotonergic system may underlie the sex-specific behavioural phenotypes exhibited by fish raised from embryos exposed to venlafaxine. It is also possible that venlafaxine perturbs sex-specific hormonal levels, as fluoxetine has previously been shown to alter oestradiol levels in goldfish [43]. Together, these studies point to a possible role for venlafaxine-mediated alterations in the development of the serotonergic system and target tissue responsiveness as a possible mechanism for its sex-specific effects on behaviour in zebrafish.

An interesting finding from this study was the limitation in the aerobic scope of female fish following venlafaxine deposition. Aerobic scope, or the increase in aerobic metabolic capacity an animal can exhibit, has been used to describe the limit of oxygen delivery to the mitochondria in tissues [44]. Limitations in this capacity may pose detrimental consequences to the animal and the energy required for activities, including growth and reproduction, may be compromised [45]. The discovery that this developmental effect of venlafaxine is female-specific is novel and may be associated with the development of the monoaminergic system, which has been shown to be sex-specific (see above). For instance, clear differences in neuroanatomy and signalling in noradrenergic systems have been observed, with neurons of the locus coeruleus–norepinephrine system in females being longer and in greater numbers relative to male conspecifics [46]—also exhibiting greater activation of this system [47]. Additionally, the serotonergic system is also sexually dimorphic, with increased levels of serotonin production in males relative to females in the human brain [35,36]. Additionally, serotonin transporter function was higher in the vasculature of female compared to male rats [39,48]. Aerobic metabolism is in part driven by cardiac performance and vascular blood flow capacity [49,50]; however, catecholamines also play a critical role in supporting oxygen availability to aerobic muscles [51]. The reduced aerobic scope seen only in females from venlafaxine may suggest disruptions in the catecholaminergic system. This is possible as females and males exhibit different responsiveness to noradrenaline in the cardiac tissue [52]. However, our findings may also be an indicator of a hormonal disturbance in the animal, as despite a diminishment in the aerobic scope of the animal, we do not see a clear effect on animal performance. Aerobic scope is typically significantly correlated with the maximum swimming capacity of the animal [53]. However, despite a marked reduction in the active metabolic rate and aerobic scope seen with venlafaxine, there is no obvious impact on U_Crit_. Studies have demonstrated the positive relationship between metabolic rates and cortisol levels, with metabolism increasing with higher cortisol levels in fish [54,55]. We have previously shown that adult zebrafish exposed to venlafaxine at the zygotic stage, particularly females, exhibit attenuated cortisol stress responses [10]. We ruled out limitations in cortisol synthesis in that study, suggesting perturbations at the level of hypothalamus–pituitary functioning [10]. As U_Crit_ did not change in our assessment, it remains to be seen if this is indicative of perturbed cortisol responses in female fish, rather than limitations in cardiac performance. To obtain aerobic scope, animals underwent a similar air exposure stressor as implemented previously, where venlafaxine-treated females were unable to produce cortisol levels to the same level as males [10]. Perhaps the inability to properly respond to stress diminished the concomitant increase in aerobic capacity. Further testing will be required to determine the specific mode of action underscoring the diminished aerobic scope in fish exposed to venlafaxine at the zygotic stage. As we have previously noted multiple generations of impact in regard to the stress response [10], it may be prudent to investigate whether similar disruptions to metabolic capacity are transferred across generations.

In summary, our results show that early-life exposure to venlafaxine may disrupt the metabolic and behavioural performance of zebrafish across their ontogeny. Zygotic exposure to this antidepressant leads to reductions in aerobic scope while increasing the boldness of juvenile fish. Impacts are sex-specific, supporting our previous findings of female-specific stress response disruption [10], and the observation that males experience behavioural alterations at the adult stage. While we did not determine the specific mechanisms of action behind these sexually dimorphic responses, we propose disruptions to the underlying monoaminergic systems of these animals as a possible reason for these sex-specific effects. As a selective serotonin and norepinephrine reuptake inhibitor, the dual mechanism of action of venlafaxine increases its potential targets of disruption. These results lend credence to the idea that monoaminergic systems may be sexually dimorphic in fish. Importantly, early-life exposure to venlafaxine may impact the developmental programming of metabolic and behavioural performance, leading to sex-specific consequences in fish.

## Figures and Tables

**Figure 1 biology-11-00250-f001:**
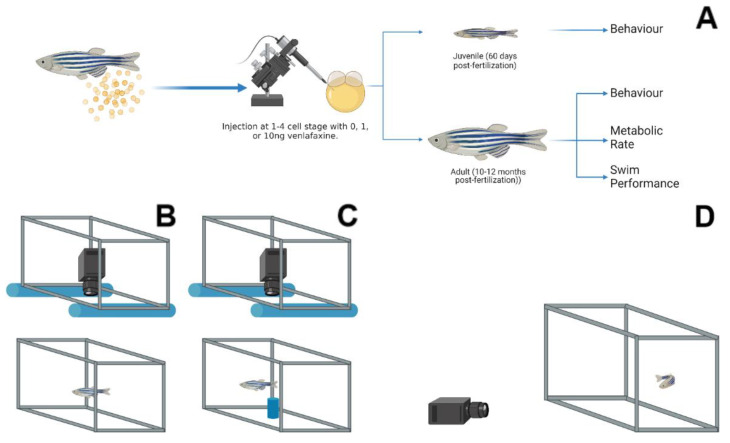
Experimental design and behavioural assessments. (**A**) Zebrafish eggs were collected immediately after fertilization and microinjected with either 0, 1 or 10 ng venlafaxine. Eggs were raised to either 60 dpf, and assessed for behaviour, or until 10–12 months post-fertilization and assessed for behaviour, metabolic rate, and swim performance. A top-down camera view was utilized for Novel tank (**B**) and Novel object (**C**) assessments. An air stone (blue) was the novel object (**C**) placed in the aquarium. (**D**) Lateral view behavioural assessment was carried out with the camera placed on the same surface of the tank of interest (**D**). The images were created with BioRender.com.

**Figure 2 biology-11-00250-f002:**
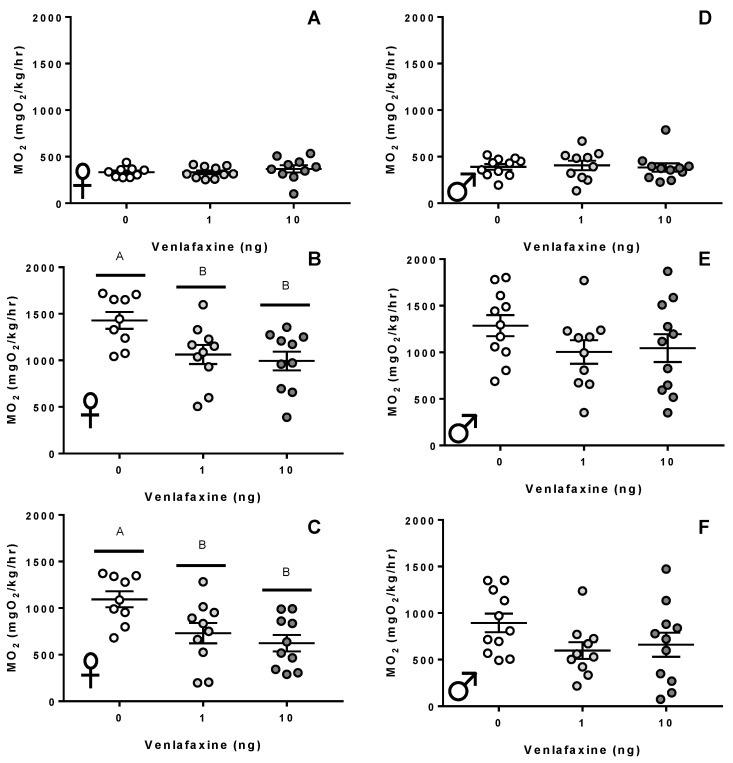
Metabolic rate. (**A**,**D**) Resting metabolic rate, (**B**,**E**) active metabolic rate, (**C**,**F**) the aerobic scope of 10–12 month old female (**A**–**C**) and male (**D**–**F**) zebrafish exposed to either 0, 1, or 10 ng venlafaxine at the 1–4 cell stage (*n* = 10–15 for respiration measurements). Different letters denote significant differences between groups. Individual data points and mean are shown for each graph (±S.E.M).

**Figure 3 biology-11-00250-f003:**
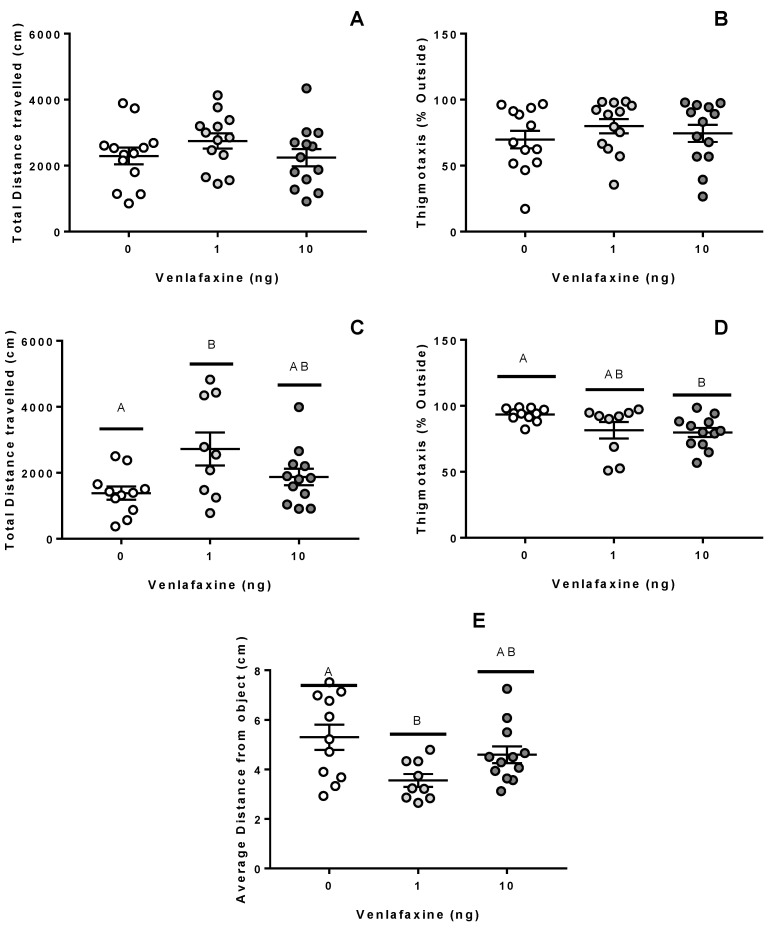
Behavioural assessment of juvenile fish. Behavioural responses of juvenile fish in lateral view (**A**,**B**), open field (**C**,**D**) or novel object assessments (**E**). Zebrafish were originally exposed to 0, 1, or 10 ng venlafaxine at the 1–4 cell stage. All assessments were recorded for 5 min. The novel object assessment occurred immediately after the open field assessment by the addition of an air stone directly into the middle of the arena (*n* = 9–13). Different letters denote significant differences between groups.

**Figure 4 biology-11-00250-f004:**
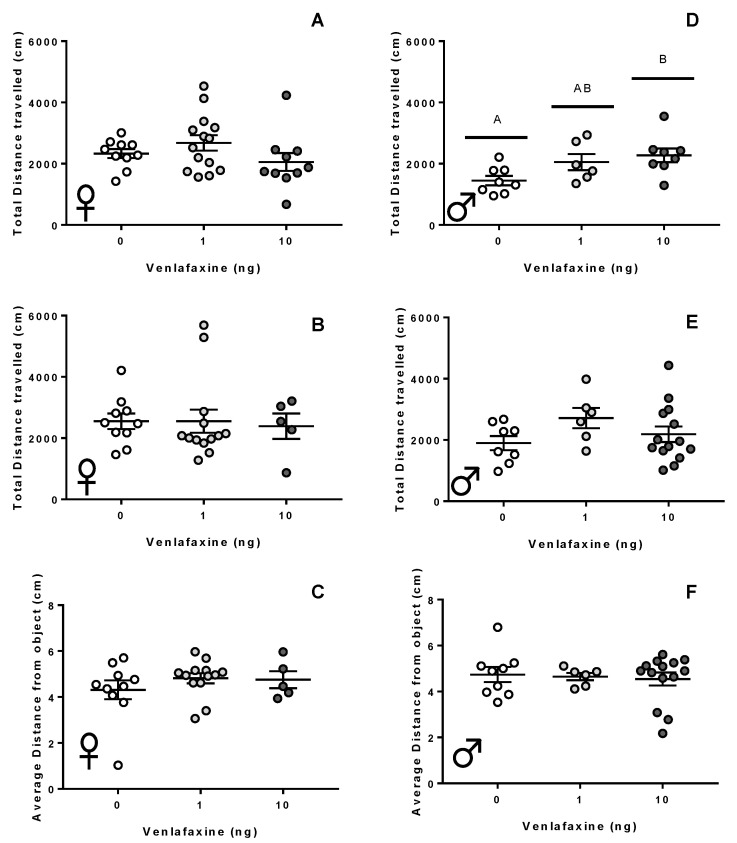
Behavioural assessment of adult fish. Total distance travelled by adult female (**A**–**C**) or male (**D**–**F**) zebrafish observed in a lateral view (**A**,**D**), open field (**B**,**E**), or novel object assessments (**C**,**F**). Zebrafish were originally exposed to either 0, 1, or 10 ng venlafaxine at the 1–4 cell stage. Zebrafish were recorded for 5 min immediately after placement into the arena. Total movement and the total distance away from the novel object are shown. Individual data points are shown with a mean ± SEM, with different letters used to denote significant differences between means.

## Data Availability

Data provided as a .xlsx file with the Appendix A.

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
