# Peer review of "Sex-Specific and Long-Term Impacts of Early-Life Venlafaxine Exposure in Zebrafish"

_biology, 2022, doi:10.3390/biology11020250_

Round 1
Reviewer 1 Report
The study looks at the effects of the antidepressant venlafaxine on zebrafish, investigating the effects of microinjection at the 1-4 cell stage on adult metabolism and behaviour. The authors have previously looked at other effects of venlafaxine on zebrafish and this work therefore builds on an existing body of information in this area.
Main comments:
Lines 75-77. I would like to see a bit more information on how the fish were raised. Twelve independent clutches were used – were there embryos injected with each treatment from each clutch? How many eggs were injected per treatment? What was the survival rate, were embryos from different treatments raised in different tanks – how many per treatment, were they raised at the same density? Was the mortality the same for each treatment? I don’t really get a feel for the size/numbers of animals involved from this introductory paragraph. It is then unclear whether the same individuals were used for respirometry measurements and behavioural measurements. Given stocking density and repeated handling can affect behaviour in particular, I think it is important to have a bit more clarity here on how the fish were treated. Line 120 mentions ‘uneven contribution of replicates from each tank’ due to survivability of individual treatments and random sex ratio between tanks. It isn’t clear to me if treatment affected either parameter. I think more details are needed.
Minor comments:
Line 77 is a repeat of line 74.
Line 88: Data were…
Line 132: data are shown
Line 219: Typo- possible
Reviewer 2 Report
The paper is interesting and the material and methods are sound. The group of investigation has a good experience in the topic. Moreover, they published previous articles using Venlafaxine in zebrafish. But before publication, the authors should add morphological analysis supporting their findings with the behaviour test demonstrating better the sex-specific changes.
Reviewer 3 Report
The manuscript in question seeks to bring a new way of analyzing the effects of venlafaxine in long-term exposure in zebrafish. The proposed methodology is innovative and aims to bring a more effective way of assessing the damage that venlafaxine causes in behavior and metabolism. However, the author’s sin on some points as can be seen in my considerations below.
- It is necessary to explain why the choice of the dose injected into the embryos, this justification is not clear in the text.
- Phenol red using to confirm the injection, could it not influence future tests and interact with the applied venlafaxine?
- In the methodology, it is not clear how the randomization in the choice of fish, and how was made the blinding in treatments.
- To improve the understanding and visualization of the experimental design and on how the behavioral tests were performed, I suggest that figures and diagrams be made.
Round 2
Reviewer 2 Report
The manuscript is now suitable for publication.